# When the going gets tough, the entrepreneurs get less entrepreneurial?

**Joeri van Hugten**[1]*, **Johanna Vanderstraeten**[2], **Arjen van Witteloostuijn**[3], **Wim Coreynen**[2,4]

1 Management & Organization, School of Business and Economics, Vrije Universiteit Amsterdam, Amsterdam, The Netherlands, 2 Department of Management, Faculty of Business and Economics, University of Antwerp, Antwerp, Belgium, 3 Ethics, Governance and Society, School of Business and Economics, Vrije Universiteit Amsterdam, Amsterdam, The Netherlands, 4 School of Management, Zhejiang University, Hangzhou, China

* j.g.w.j.van.hugten@vu.nl

**Data Availability Statement:** Availability of codes, data and material: An abbreviated R script is in the appendix. Data cannot be shared publicly because, as part of our ethics statement and consent agreement with participating entrepreneurs we

## Abstract

We investigate how the 2008–2012 economic crisis relates to entrepreneurs' confidence ten years later and, in turn, their venture's entrepreneurial orientation. Conceptually, we introduce the new concept of 'hard times' to capture an entrepreneur's sense of their venture's hardship during the crisis. Theoretically, we extend ideas on imprinting, to build the argument that hard times cause a persistent reduction in an entrepreneur's entrepreneurial self-efficacy and sense of job security, both of which in turn cause a contemporaneous reduction in their venture's entrepreneurial orientation. We contrast the crisis imprinting hypothesis with a hypothesis from the more established behavioral theory of the firm. Strikingly, rich data of about 300 Flemish entrepreneurs and their ventures are in line with a small crisis imprinting effect.

## Introduction

We study how a crisis can cause a long-term loss in entrepreneurs' confidence which in turn reduces the entrepreneurialness of their small or medium-sized enterprise (SME). There is a clear need to understand more about crises, and especially the long-term consequences of crises, as we experience the inflationary effects of the Russo-Ukrainian war, next to the economic consequences of the covid pandemic, only ten years after the last financial meltdown [1–3].

Beyond the importance of studying crises per se, the effect of crises on SME entrepreneurialness is of special relevance because being entrepreneurial may be a way to overcome, or navigate through, crises. For example, non-entrepreneurial SMEs may have savings that ensure stability during a small dip, but they may be unable to adapt to the environment after a greater shift (e.g., a crisis) [4–6], which ultimately leads to venture collapse. So, if crises reduce entrepreneurialness, then overcoming crises may be less likely, and the harmful effects of crises may be perpetuated in a vicious cycle. Conversely, being entrepreneurial may help SMEs adapt, and as the backbone of a country's economy [7], that may help the whole economy recover. To better understand this relationship, we contribute an analysis of uniquely rich data that combines the entrepreneur- and SME-levels.

have promised to limit access to their responses to only those people involved in the project. This data collection is still ongoing. Part of the data underlying the results presented in the study might be available in the future from UNIZO. Our survey is administered in cooperation with a Belgian cooperative of small businesses (UNIZO). They can be contacted here: UNIZO vzw Willebroekkaai 37 1000 Brussel https://www.unizo.be/contact They could give access to the survey part of the data we used in this paper if one refers to our joint project (it has been named 'Ambitie in Ondernemen' 'Road to Growth' and 'Strategie en Groei'). Our financial information (e.g., profit and firm size) comes from a private third party organization: GraydonCreditsafe Belgium Uitbreidingstraat 84/b1 2600 Antwerpen (Berchem) https://graydon.be/nl/contact For a fee, part of the data underlying the results presented in the study are available from them. The dataset is also stored on a onedrive at the University of Antwerp. Researchers interested in reproducing the study could also request access there. The University of Antwerp supports FAIR and open science. For this study we have used the data up to what we call 'the second wave'. A random cut off by us as researchers to make one dataset with the first 532 respondents. If future researchers request this data, they may use the updated information which might cause some differences in results, although recently we have stopped asking about the 2008-2012 crisis.

**Funding:** The author(s) received no specific funding for this work.

**Competing interests:** The authors have declared that no competing interests exist.

Furthermore, by studying confidence in relation to crises, we want to bring attention to loss of confidence in entrepreneurship research. That is important because the dominant portrayal of entrepreneurs as overconfident nudges us away from a more humanized, empathetic stance; one that suits a study on normal, everyday entrepreneurs [8]. Moreover, to the extent that confidence mediates between crises and entrepreneurialness, this study may identify confidence as a lever for policy, consulting, and training to build venture crisis resilience [9, 10], and, indirectly, macro-economic resilience.

We conceptualize crisis as an entrepreneur's retrospective sense of their venture's hardship during the financial crisis, captured in a new concept we call 'hard times'. We conceptualize entrepreneurialness using the established concept of entrepreneurial orientation (EO) [11, 12]. So, being entrepreneurial refers to ventures being innovative, proactive, and risk-taking. We conceptualize confidence in two ways: entrepreneurial self-efficacy (ESE) [13] and sense of job security (SJS) [14]. We agree with Bullough et al. [10] that ESE is an interesting concept to understand entrepreneurs in times of adversity. Furthermore, SJS is especially interesting for entrepreneurs, because the confidence in keeping their job is directly tied to the confidence in their ventures' survival (as opposed to being fired by another person). In sum, concretely, we study to what extent entrepreneurs who today (i.e., 2017–2019) perceive the 2008–2012 financial crisis to have been harder for their venture, still have lower ESE and SJS today, and therefore have ventures with lower EO today.

Theoretically, we extend ideas on imprinting [15, 16], to build the argument that hard times imprint on entrepreneurs' cognition and affect and thereby cause a persistent reduction in EO. We refer to this argument as the crisis imprinting theory (CIT) hypothesis.

This paper contributes the concept of 'hard times' to entrepreneurship research on crises [1–3, 9]. This concept's subjective nature helps recognize heterogeneous experiences of crises and helps capture the impact of crises more proximately to subjective consequences of interest (e.g., confidence). Subjective measures are used in crisis research broadly (e.g., [10, 17]), but the concept of 'hard times' is uniquely valuable to entrepreneurship research because it frames a person's crisis experience in connection to their venture.

This paper contributes to the growing interest in antecedents of EO [18]. To the extent that the concept of hard times overlaps with past venture performance, CIT reverses the usually theorized causal direction, to view performance as antecedent of EO [4, 5, 19]. The most related studies are on contemporaneous, industry-level performance as an antecedent of EO [20, 21]. For example, EO may be higher during growth in industry-average sales [22]. The contribution relative to that work is to focus on long-term and entrepreneur-level antecedents (i.e., hard times).

This paper contributes to the recently emerging entrepreneurship literature on performance feedback (i.e., performance as an antecedent rather than outcome) [23]. The entrepreneurship context is relevant because it makes decision-makers' individual-level characteristics more impactful. For instance, an SME's response to performance feedback may be mediated by individual-level concepts in a way that would not be considered by theories focused on large firms [24]. By developing CIT we contribute exactly such an argument. In addition, we actually measure the entrepreneurs' cognition and affect that are usually only theorized as mechanisms [24, 25].

The established theorization of performance feedback is below-aspiration performance [26]. If contrasted to another theorization, that other theory is usually the threat-rigidity hypothesis [27]. Among entrepreneurship studies, that contrast has been made once in the context of past performance driving or impeding business model change [25]. In addition, Kreiser et al. [22] use the threat-rigidity argument on its own to connect environmental hostility to EO. In this paper, we contrast the below-aspiration performance theorization to CIT.

Relative to the more familiar contrast to threat-rigidity, the contrast to CIT reveals that long-term consequences cannot be simply extrapolated from theories on the short-term consequences.

## Crisis imprinting theory

When setting out to study the long-term consequences of the financial crisis, we noticed that we needed a new construct to focus the overwhelming and elusive phenomenon of 'an economic crisis' into something more concrete and tangible. Conceptualizing crisis using Staw et al.'s [27] concept of 'a threat' would emphasize crisis' immediate consequences (e.g., [17]). That emphasis does not fit with our interest in long-term consequences that occur after the threat has (long) passed. The most closely related existing concept that we know of is 'below-aspiration performance' within the behavioral theory of the firm [26]. However, we feel, and we will show through contrasting later, that that concept does not explain the sense of a *force majeure* and affect specific to crises. In its stead, we introduce 'hard times' as a new concept that can fulfill that role in our study of SMEs in times of economic adversity.

At the construct level, we define hard times as an entrepreneur's degree of dissatisfaction with their venture's performance (e.g., in terms of profit or number of employees) in crisis times. That construct reflects a number conceptual moves based on our intuition about what may be meaningful. First, we intuit that it may be meaningful to consider between-entrepreneur heterogeneity in the experience of crises, as opposed to, for instance, an indicator variable for 'this industry-year has a crisis' (e.g., [2]) or the inverse of growth in industry-average sales [22].

Second, we intuit that it may be meaningful to consider an entrepreneur's perception of reality rather than reality per se, as that should be more proximately tied to their cognitions, attitudes, beliefs, emotions, affects, and behaviors (e.g., [28–30]). Thus, hard times refers to subjective performance satisfaction rather than objective performance data. This also means that hard times is an entrepreneur-level concept because the entrepreneur should not indicate venture performance, but instead indicates their satisfaction with venture performance.

Third, we intuit that the venture's performance is a meaningful unit to evaluate the hardship. By contrast, the entrepreneur's satisfaction with their personal finances or their country's inequality or unemployment are less intuitively connected to cognitions and affects that might later shape the venture's EO. Furthermore, satisfaction with venture behavior or strategy during the crisis may affect EO via more direct mechanisms than crisis imprinting.

Fourth, we focus on variation in the degree of *dis*satisfaction. We intuit that the range 'very dissatisfied–dissatisfied–somewhat dissatisfied–neutral' is the interesting and relevant range when studying crises.

Conceptualizing crisis as hard times is the 'crisis' part of crisis imprinting theory (CIT). The 'imprinting' part is using imprinting [15] to theorize the persistence of hard times' impact such that entrepreneurs today still differ based on their experience during a past crisis. Marquis and Tilcsik [16] distinguish three characteristic aspects of imprinting. First, the 'when'; "a sensitive period characterized by high susceptibility to environmental influence" (p. 199). We propose that, a crisis is such a sensitive period where all entrepreneurs' affects and cognitions are highly susceptible to environmental influence. Crises are periods of uncertainty, which heightens susceptibility to environmental influence [16]. Cognitions and affective reactions can become ingrained, but crises can shake up those foundations. For example, when people see usually stable banks or even entire countries go bankrupt, they are triggered to fundamentally reconsider their beliefs. On this aspect, CIT's approach is different from the more common approach to define sensitive periods using the characteristics of the bearer of the imprint

(e.g., using 'early career' as the window where a manager is sensitive to be imprinted by recessions [31]). By contrast, CIT considers crises to be so extremely impactful that even veteran entrepreneurs are susceptible to them. But, low age or tenure may strengthen the imprint.

Second, the 'what'; the "stamp of the environment" (p. 200). An economic crisis is an environment with unique content to imprint. For instance, hard times can imprint the content that the venture is fragile [3], and that factors outside of the entrepreneur's control can cause venture failure [32]. The imprinted content becomes clearer in the context of a concrete affect or cognition that bears the imprint, as discussed in the hypothesis section.

Third, the 'how long'; i.e., the "persistence of imprints" (p. 201). At the individual level, imprints persist mostly because of the low susceptibility to counteracting influences after the imprint [16]. In non-crisis times, environmental influences are not powerful enough to unsettle the ingrained cognitions and affects [33]. In other words, people are not so 'impressed' by environmental influences in non-crisis times.

In sum, similar to how professionals develop persistently different work routines depending upon organizational munificence at the time of hiring [34], or how CEOs take less risk if they started their managerial career during a recession [31], entrepreneurs can have a persistently different cognitions and affects depending upon the degree of hardship experienced during a crisis.

## Hypotheses

The above theory suggests that entrepreneurs are susceptible to an imprint from hard times that has a persistent effect. This section discusses the auxiliary assumptions that connect that imprint to EO via entrepreneurs' confidence. We conceptualize such confidence in two ways: entrepreneurial self-efficacy (ESE) and sense of job security (SJS). ESE is the entrepreneurship-specific version of the broader concept of self-efficacy [35]. It has a cognitive component in the self-estimation of the likelihood of successfully completing a task (in this case, entrepreneurial tasks), as well as an affective component [13]. SJS also captures both the cognition-based estimation of the likelihood of job loss, as well as the affect 'security' that a respondent associates with that likelihood [36, 37].

We connect hard times to ESE with two premises. First, hard times imprint content onto ESE that is similar to the effect of perceived low venture performance on ESE. Second, perceived low venture performance influences ESE. To support this second premise, we would like to use the evidence that venture performance has been correlated with ESE, but existing evidence only considers ESE as an antecedent to performance [38]. Thus, to bridge that gap we build a syllogism on two sub-premises: 1) evidence on the effect of prior individual performance on self-efficacy generalizes to an effect of perceived prior venture performance on ESE, and 2) there is evidence that prior individual performance positively relates to self-efficacy [35]. Specifically, for a between-person analysis (like we do), a meta-analysis that included various kinds of self-efficacy reveals a .42 correlation [39]. Moreover, in a lab setting, the effect of an experience of low performance on self-efficacy persists even after performance recovery [40].

Now we can also further support the first premise. We use the sub-premises; 1) the imprint of hard times is similar to a contemporaneous effect of hard times, and 2) the contemporaneous effect of hard times is similar to the effect of low prior performance (i.e., reduce ESE). The most related evidence for this second sub-premise is that entrepreneurs' perceived threat of war violence against them indeed reduces ESE ($r = -.23$) [10].

With these auxiliary premises we deduce from CIT:

Hypothesis 1 (H1): Hard times persistently decrease entrepreneurial self-efficacy.

We use a similar premise setup to deduce hypothesis 2. First, hard times imprint content onto SJS that is similar to the effect of economic conditions and firm performance on SJS. Second, economic conditions and firm performance influence SJS. To support this second premise, we would like to use the evidence that economic conditions and firm performance have been correlated with entrepreneurs' SJS, but existing evidence only considers SJS among employees. So we again use the sub-premise that such evidence generalizes. And the evidence across employee samples suggests that economic conditions do relate to job security [14], and in particular that firm performance positively relates to job security (r = -.28) [37]. This relationship may be even stronger for subjective performance measures such as hard times. For example, the mere expectation of downsizing reduces job security (r = -.29) [41].

Regarding the first premise, we believe it is a conservative premise because a crisis may impact SJS more strongly than economic condition or firm performance. What is not yet evident is whether such an effect persists. CIT implies that it does, and we believe that is a plausible premise; hard times could imprint the content that the venture is fragile in a way that persistently makes the threat of downsizing somewhat salient.

In all, with these auxiliary premise, we deduce from CIT:

<u>Hypothesis 2 (H2)</u>: Hard times persistently decrease sense of job security.

The effects of ESE and SJS on EO, which we need to complete our mediation logic, are not the focus of CIT. However, to set up the mediation credibly, we want to make those relationships explicit in the form of a pair of expectations. Note that we call these pairs "expectations", and not "hypotheses", because they are based on the most related empirical (correlational) evidence, not derived from theoretical arguments.

First off, in a study that used ESE and EO as separate independent variables, measured with the same scales we use, the correlation between them was .32 [19]. We can also connect ESE and EO via other constructs. For instance, via promotion focus. ESE is strongly and positively correlated (r = .60) with promotion focus (and uncorrelated with prevention focus) [42]. Promotion focus is, in turn, positively correlated with EO [43]: Specifically, the correlation between EO and three types of promotion focus were .27, .43, and .52, respectively. Therefore, ESE can be expected to positively relate to EO. Specifically, this argument suggests a correlation in the range between .60*.27 = .17 and .60*.52 = .31. Another possible intermediate construct is improvisational behavior. ESE is positively correlated (r = .49) with improvisational behavior at work by new venture founders [44]. Improvisational behavior is related to EO. For example, EO is associated with introducing new product lines as opposed to tried-and-true products [11], while improvisational behavior involves "finding new uses for existing methods or equipment" [44]. If entrepreneurs apply improvisational behavior at the strategic level, that would increase the venture's EO. In sum, we expect:

<u>*Expectation 1 (E1)*</u>: *Entrepreneurial self-efficacy increases entrepreneurial orientation.*

To connect SJS to EO we can use creativity and innovative work behavior as intermediate constructs. SJS is positively related to creativity (dichotomous variable effect size = .17) [45] and innovative work behavior (r = .12 and .16) [46] among employees. In turn, creativity is positively related to EO (likert scale effect size = .31) [47], and innovative work behavior correlates between r = .53 and r = .64 with the three subdimensions of EO [48]. In sum, we expect:

<u>Expectation 2 (E2)</u>: Sense of job security increases entrepreneurial orientation.

H1 and E1 together imply the hypothesis that hard times affect EO via ESE, and H2 and E2 together imply the hypothesis that hard times affect EO via SJS.

<u>Hypothesis 3 (H3)</u>: Hard times persistently decrease entrepreneurial orientation via entrepreneurial self-efficacy.

<u>Hypothesis 4 (H4)</u>: Hard times persistently decrease entrepreneurial orientation via sense of job security.

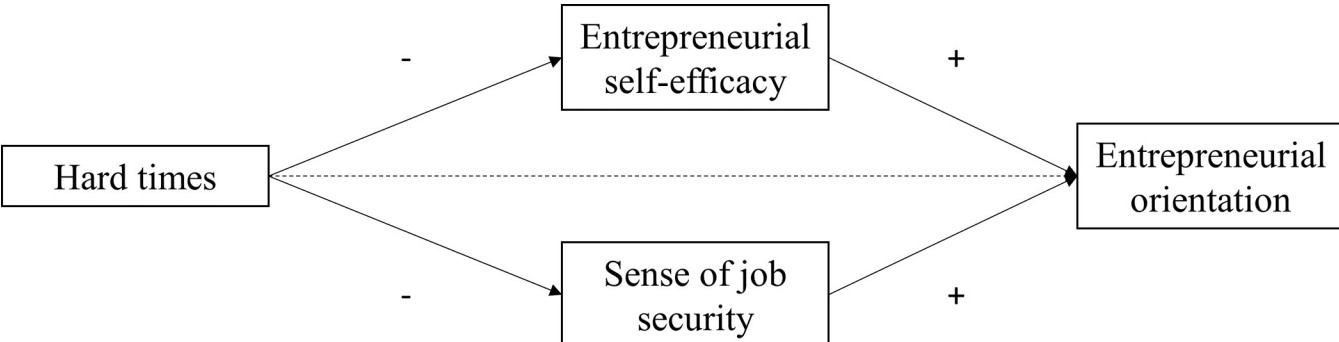

**Fig 1. Crisis imprinting theory's hypotheses and expectations overview.** The dashed line indicates that both a negative (H5) and a positive effect (H5alt) are hypothesized.

Taking H3 and H4 together, we can close the circle of our mediation logic:

Hypothesis 5 (H5): Hard times persistently decrease entrepreneurial orientation.

We specify H5 because it is rejected or supported by different evidence than H3-4, depending on the presence or absence of counteracting mechanisms in addition to the mechanism of H3-4. H5 is interesting exactly because there is a theory that predicts such a counteracting mechanism. Specifically, the data conceptualized as 'hard times' (i.e., extent of dissatisfaction with performance in times of crisis) might alternatively be interpreted as a measure of below-aspiration performance. The behavioral theory of the firm suggests that firms stay on course unless prompted to change by problems. Firm performance becomes a problem when it is below the firm's aspiration level (e.g., high dissatisfaction). Thus, below-aspiration performance prompts search and risk-taking [26]. Search and risk-taking are aspects of EO [11]. Hence, the behavioral theory of the firm predicts hard times increase rather than decreases EO. That provides a meaningful baseline H5alt:

*Hypothesis 5alt (H5alt)*: *Hard times increase entrepreneurial orientation.*

Fig 1 summarizes the hypotheses. The dashed line indicates that both a negative and a positive effect are hypothesized.

## Methods

As part of an ongoing consulting project involving entrepreneurs and their SMEs, we asked 532 entrepreneurs in great detail about their venture, as an enterprise, and about themselves as an individual, a couple of weeks later. In the venture questionnaire, we measured EO and satisfaction with performance. In the individual survey, we measured our pair of mediators: ESE and SJS. Thus, measures of concepts directly linked in our conceptual model are never sourced from the same survey. This time lag substantially reduces the threat of common-method variance, probably to (close to) zero [49]. The sample contains entrepreneurs and their ventures from Flanders (the Dutch-speaking part of Belgium), from 55 three-digit industries. Of the observations for which the data is complete, all but six ventures have fewer than 50 employees, and 104 even have fewer than five employees.

The consulting project was run by the association for small and medium-sized businesses in Flanders (known as UNIZO), co-financed by the Flemish Agency for Innovation and Entrepreneurship, and backed by a university research team and a group of professional coaches. For a more comprehensive description of the project, see [50]. The project's aim was to provide strategic advice to SMEs regarding how they could improve their performance. Ventures were approached via UNIZO and coaches' professional networks. Entrepreneurs applied for

an intake conversation in which they indicated their written consent to join or not (the interviewers also provided options to join other trainings instead of our consulting project). Then, they went, step by step, through a series of measurement exercises followed by meetings with coaches. Our project attracted mostly small businesses, both young and old, that sought advice on a variety of topics, such as dealing with threats and decline, how to exploit an opportunity they were seeing, or how to become more innovative. No approval was obtained from an ethics committee at the start of the project because the study was conceptualized before ethics committees became the norm, but the study passes the 2023 check for ethics of Research Ethical Review Board of the School of Business and Economics, Vrije Universiteit Amsterdam.

This sampling procedure's integration with government-promoted coaching has the advantage of broad appeal and legitimacy over normal surveys. A disadvantage is that ventures that were not looking for advice selected out of participating. To estimate the extent of selection bias, we can look at ventures that volunteered to apply compared to ventures that were recruited to apply (about 50/50 in the sample). S1 File shows how our sample appears representative in terms of most variables except for relatively low EO in our sample. This may mean that the range of the dependent variable is smaller than in the population. The main consequence would be that effect sizes we estimate may be smaller and thus more conservative.

Our sample only contains ventures that have survived the financial crisis. Often such survival implies a bias. It would imply a bias for this paper if the theory we test was about the effect of the crisis in general. However, the theory we test is about the effect of the crisis on the survivors, many years after their survival. Therefore, our sample is the relevant set of observations (as opposed to being a subset of ventures 'treated with crisis'). The remaining set of observations is only subject to the selection bias discussed above.

*Hard times* is measured as the dissatisfaction with venture performance. We consider four aspects of hard times: 1a) dissatisfaction in terms of profit during the crisis, 1b) dissatisfaction in terms of the number of employees during the crisis, 2a) dissatisfaction in terms of profit for some time after the crisis, and 2b) dissatisfaction in terms of the number of employees for some time after the crisis. For the SMEs we study, hard times occurred in the context of the 2008 global financial crisis. In Europe, the financial crisis led to a domino effect causing the Eurozone debt crisis first in Greece and later in Ireland and Portugal. This deepened the concerns well into 2011. In Belgium, a 15-year stretch of around 2% growth in GDP per capita, with even 3% growth in 2008, abruptly turned to a -3% recession in 2009. Positive GDP growth returned in 2010, but then three years of almost 0% growth followed [51]. Regarding unemployment, a declining trend was broken in 2009 by a sharp increase that lasted, barring a small recovery in 2011, well into 2015. Since then, Belgium showed a stable 1% growth rate with steadily declining unemployment until the 2020 pandemic [52]. It is in regards to those periods of impact that we asked entrepreneurs to evaluate their dissatisfaction during crisis times as well as recovery times. This evaluation happens in our survey in 2017–2019. That timing is noteworthy in two ways. First, it means that there is a five-to-ten year lag between the experience of hard times and the rating of it, creating room for hindsight bias. We address this concern using covariates (see below for more detail). Second, it gives a more concrete idea of time scale of the persistence we theorize. Specifically, hard times should affect EO five to ten years later. This time scale is in line with prior studies of economic situation imprints on individuals, which use a time scale of an average of five [34] or 22 years [31].

In line with spline models common in performance feedback studies [23], we cluster all neutral and satisfied entrepreneurs together at pole 4, because we believe that the degree of *dis*satisfaction is the relevant range. Then, for the sake of interpretation, we reverse code the responses such that higher values reflect harder times. Finally, we consider not only hard crisis times, which refers to performance in the midst of the crisis, but also hard recovery times,

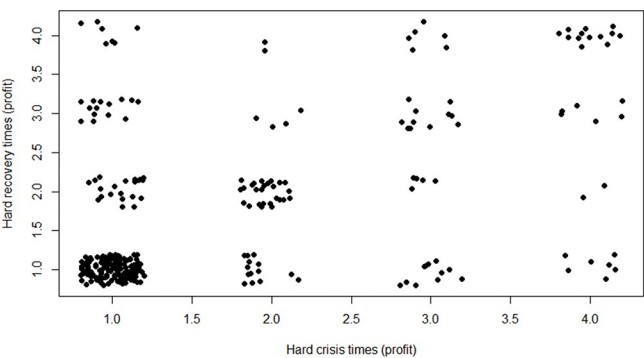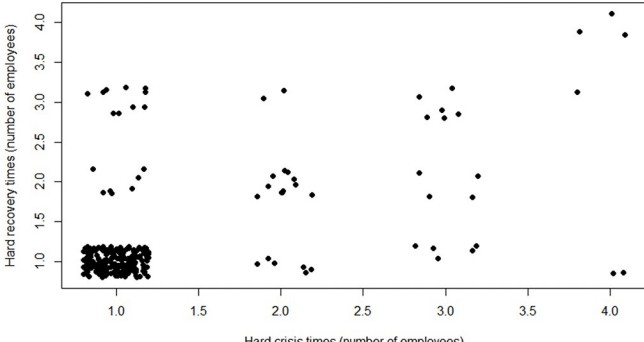

**Fig 2. Scatterplot of hard crisis times compared to hard recovery times.** For profit (left), *n* = 300; for number of employees, (right), *n* = 264.

which refers to a period following the crisis. We do not impose a definition of when the crisis started or ended, but leave that to the respondents' interpretation. The final measure is a discrete scale from 1 (Neutral) to 4 (Very hard times). See S1 File for more details.

We include only ventures that responded to both during- and post-crisis questions (e.g., some did not because they were established after the crisis), reducing the *n* in our analyses from 532 to approximately 300. Fig 2 shows that quite some entrepreneurs experienced hard crisis times but low hardship afterwards, and vice versa. In terms of profit, exactly half (150 out of 300) of the ventures report no hardship during as well as after the crisis. In terms of number of employees, 207 out of 264 ventures report no hardship for both periods.

In Table 1, we list all our variables and their measures. We refer to the S1 File for more details (especially regarding factor analyses). We have the option to specify in our model a relationship between the two mediators, ESE and SJS. A cross-lagged study shows that job security and self-esteem (a concept similar to ESE) are bidirectionally related [53]. Theoretically, we do not face this two-way causality complexity, because we avoid hypotheses connecting SJS and

**Table 1. Variables and measures.**

*Entrepreneurial self-efficacy*. ESE was measured with McGee et al.'s (2009) scale. The kind of self-efficacy and the kind of performance feedback should match enough to produce large and meaningful effects. We remove three items about the 'attitude toward venturing' to reduce measurement overlap. Using the factor analysis loadings (see S1 File for details), a loading-weighted average was computed per subdimension. Then, in line with treating ESE as a formative construct, those averages were multiplied to get the ESE score.

*Sense of job security*. We measure SJS using a single item that leaves the interpretation of the concept to the respondent, as opposed to predetermining a cognitive or affective interpretation.

*Entrepreneurial orientation*. We measure EO using Covin and Slevin's [54] nine items that are divided among three subdimensions: risk-taking, innovativeness, and proactiveness [11]. The EO score equals the multiplication of the loading-weighted averages per subdimension (see S1 File for details on the factor analysis).

*Job security value*. We ask with a single survey item (1–5) to what extent the entrepreneur expects and values job security in work.

*Need for achievement*. We take the average of an 11-item scale (1–7) known as the PRF [55].

*Tenure*. In years, we have the answer to the question: How many years have you been active in this enterprise?

*Profit*. We collected objective balance sheet data regarding profit in the most recent year at the moment of responding to the survey (inverse hyperbolic sine transformed).

*Profit growth*. Profit growth is proxied by the slope of the linear time trend from a within-venture OLS regression of profit on time over the four years before participating in the study.

*Profit instability*. Profit instability is the log of the sum of squared errors from a within-venture OLS regression of profit on time over the four years before participating in the study.

*Firm size*. Firm size is measured with a categorical variable that classifies ventures into six categories according to the number of employees (1 = 1–4; 2 = 5–9; 3 = 10–19; 4 = 20–49; 5 = 50–99; 6 = 100–199).

ESE. Empirically, it turned out that modeling ESE and SJS jointly did not change their individual estimates much (available upon request). Therefore, the main results are based on the joint models.

We use *covariates* to exclude spurious correlation. First, entrepreneurs may design their venture such that they experience the level of job security they want. For example, they may opt for low risk-taking in order to have a high SJS. This implies that the entrepreneur's preference can be a common cause of EO and SJS. To capture that, we include as a covariate the extent to which the entrepreneur values job security in work. Second, entrepreneurs may design their venture such that they experience the level of ESE they prefer. To capture that, we include as a covariate entrepreneurs' need for achievement. Third, highly experienced entrepreneurs may have greater ESE and greater SJS, and they may have older, more established and thus more resilient ventures. We control for tenure to capture such noise.

Fourth, profitability could represent a common cause to extent that it reflects pre-existing differences in susceptibility to hard times, and it causes ESE and SJS, as well as EO [20–22]. It is plausible that profitability affects the concepts in our model, although most likely the different concepts are affected by profitability in different years (e.g., susceptibility to hard times may be caused by 2009 profit, while EO may be caused by 2015 profit). We address this issue by including as a covariate a venture's most recent annual profit before reporting their EO.

Fifth, we include as a covariate profit growth over the last four years. This covariate relates to the following concerns. One, entrepreneurs who are currently experiencing growth may have rose-colored hindsight bias as everything turned out well 'in the end'. In that case, high ESE and SJS now may cause a perception of less hard crisis times via rose-colored hindsight. Such rose-colored hindsight should be especially likely with high current profit as well as recent profit growth, so those covariates would capture correlation between ESE and hard times due to rose-colored hindsight bias. Two, relative hindsight bias suggests that the worse the situation now, the less hard the situation seemed in the past. So, low ESE and SJS now would cause a perception of less hard crisis times. For H5, this is more a contrasting prediction than a common cause to control for, but the profit growth covariate excludes that noise. Three, behavioral theory suggests that profit growth is negatively related to EO. Specifically, negative profit growth would reflect performance below historical aspiration. Below-aspiration performance is argued to increase risk-taking [26], and thus EO. In that case again, it is a contrasting prediction rather than a common cause, but this covariate is valuable to exclude noise for H5alt.

Sixth, larger ventures should be more buffered against hard times, so post-crisis recovery is less likely to be very dissatisfactory, and SJS should be greater. Very large ventures are often more inert, and thus less entrepreneurially oriented. However, all but one ventures in our sample have fewer than 100 employees. With that range, it may be that this covariate is not impactful.

Seventh, profit instability can be expected to reduce SJS and could be a result of risk-taking–hence, it can be a mediator in a reverse causality relation. That reverse causality suggests the opposite sign of the association between SJS and risk-taking. So, we include profit instability as a covariate to exclude an alternative explanation for H5alt, and to reduce potentially counteracting noise for H5.

The main results are outputs from structural equation models estimated using the lavaan package in R. The script is in the S1 File.

## Results

Figs 3 and 4 visualize the relationship between different types of hard times and ESE. ESE spans its entire range among entrepreneurs who did not experience hard times. By contrast,

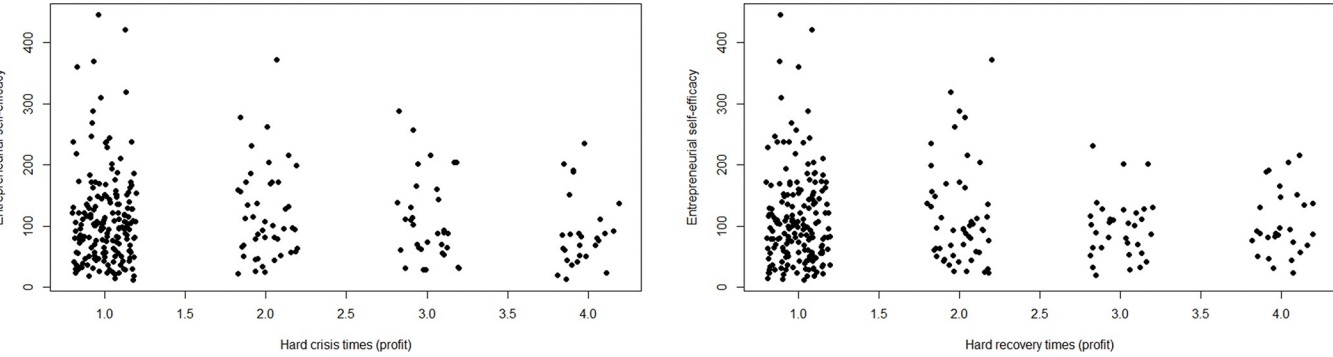

**Fig 3. Scatterplot of hard times (in terms of profit) and ESE.** *n* = 300; jitter (1) on hard times.

only low ESE occurs among entrepreneurs who experienced the hardest times. There are two high-ESE outliers in the group of entrepreneurs with hard times 3 out of 4 in terms of number of employees.

Next, Figs 5 and 6 plot the relationship between different types of hard times and SJS. When times are not hard, high SJS is far more common than low SJS. By contrast, they are almost equally common when times are hardest. When times are hardest in terms of profit while recovering from the crisis, low SJS is twice as common as high SJS.

Fig 7 shows the relationship between ESE and EO. The observations move from the origin toward the upper-right corner, but scatter diffusely, and more diffusely the farther from the origin. ESE and EO are correlated with an r = .29 (Pearsons) to r = .33 (Spearman) in line with McGee and Peterson's (2019) r = .32. In the S1 File, we report the full correlation table.

Fig 8 gives the relationship between SJS and EO. EO seems populated more evenly over its entire range at higher SJS. Conversely, high EO is rare among those entrepreneurs with low SJS. Note that the one observation with EO around 100 will drop out in the final analyses due to missing data on all employee-related variables.

This series of visuals of the raw data are all broadly in line with the hypotheses and expectations. Unusually, the figures show Y shrinking or extending in range, rather than shifting, over X. Such kinds of relationships are meaningful, but they are unfortunately underestimated by correlation-based analyses such as regression. Given that caveat, Tables 2 and 3 show the main regression results (see full model outputs in the S1 File). The inclusion of covariates generally increases the strength of the main effects slightly. One exception is that the effect of ESE on EO is twice as strong before including need for achievement.

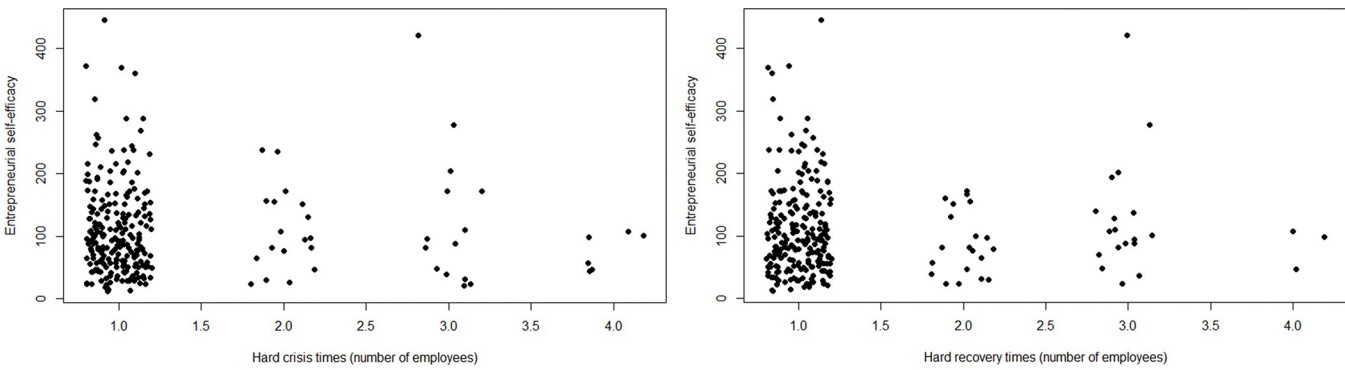

**Fig 4. Scatterplot of hard times (in terms of number of employees) and ESE.** *n* = 264; jitter (1) on hard times.

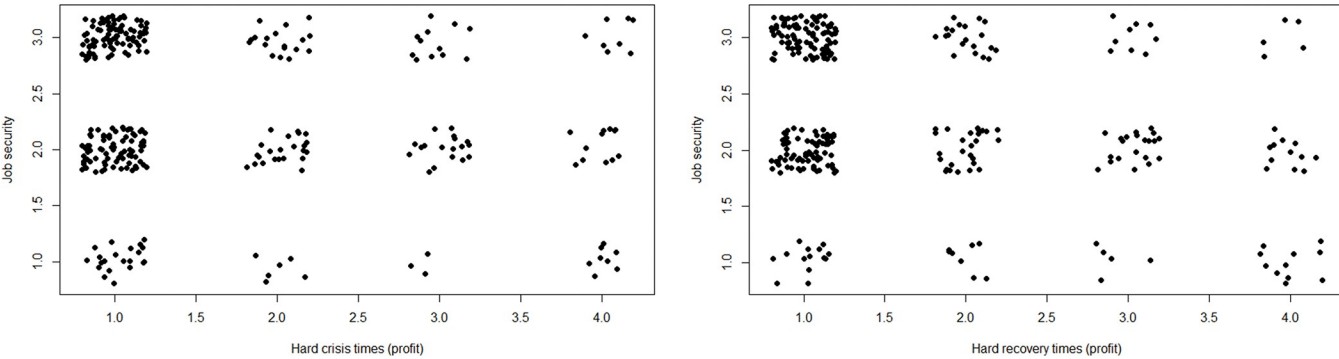

**Fig 5. Scatterplot of hard times (in terms of profit) and SJS.** $n = 300$; jitter(1) on both variables.

The main result is that hard times has a consistent negative effect on EO, in line with our crisis imprinting theory. The effect is stronger for hardship in terms of number of employees than in terms of profit. Simply put, ventures for which crisis times were 1 standard deviation harder, have 5–17% of a standard deviation lower EO today. These effect sizes are remarkably large, given the substantial time lag between the crisis and the outcome variable.

Table 2 shows that the largest indirect effect only associates a 1 standard deviation increase in hard times with a .02 standard deviation decrease in EO. Therefore, the regression supports both mediation channels (see above), it rejects the mediation hypotheses H3 and H4. We analyze the mediations in more detail later. The hypothesized mediations do explain a large part of the effect of hard times on EO, with up to 75% for hard recovery times (profit). Still, explaining a weak effect is not useful enough to consider the hypotheses supported.

Table 3 provides an overview of the two mediation paths for the four types of hard times. Looking into the components of the mediation paths, H1 is supported, albeit weakly in some cases. Some types of hardship show strong negative effects of hard times on ESE. The effect is about as strong as the effect of tenure or firm size, and three to four times stronger than the profit variables, but smaller than the contemporaneous correlation found in the meta-analysis on general self-efficacy [39]. H2 receives strong support. Especially hard times in terms of profit have a strong negative effect on SJS. Hard times' standardized effect size ranges from equally large to up to four times as large as a plausible alternative predictor: Profit growth trend over the past four years. In line with E1, ESE has a consistent, positive, and reasonably strong effect on EO. Need for achievement is the strongest predictor of EO by some distance,

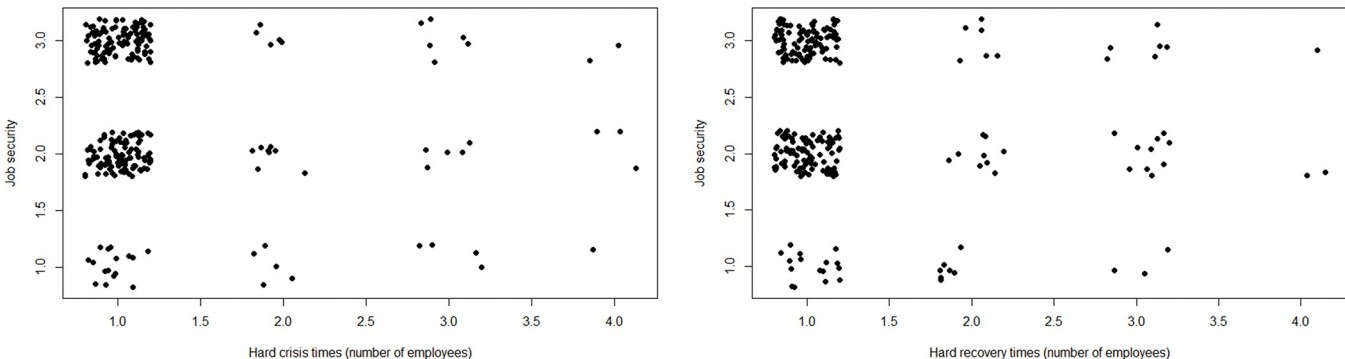

**Fig 6. Scatterplot of hard times (in terms of number of employees) and SJS.** $n = 262$; jitter(1) on both variables.

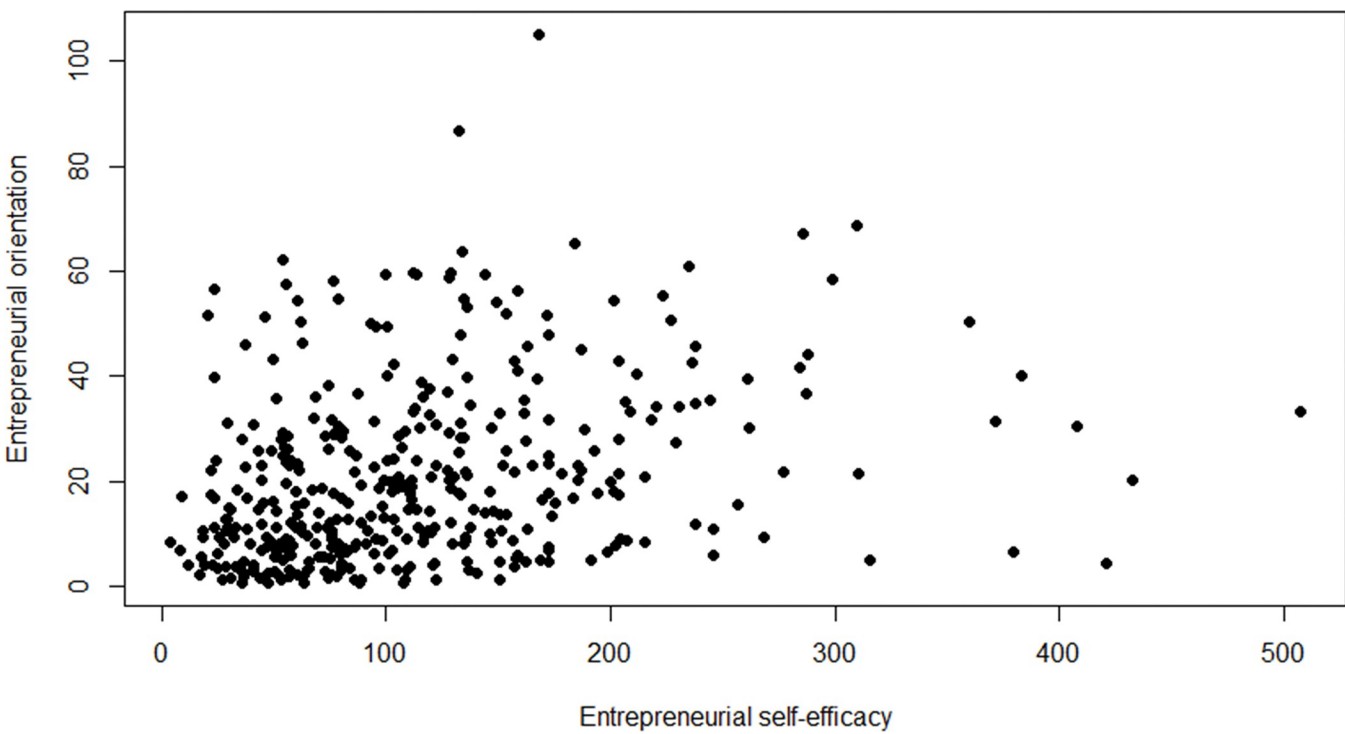

**Fig 7. Scatterplot of entrepreneurial self-efficacy and entrepreneurial orientation.** $n$ = 388.

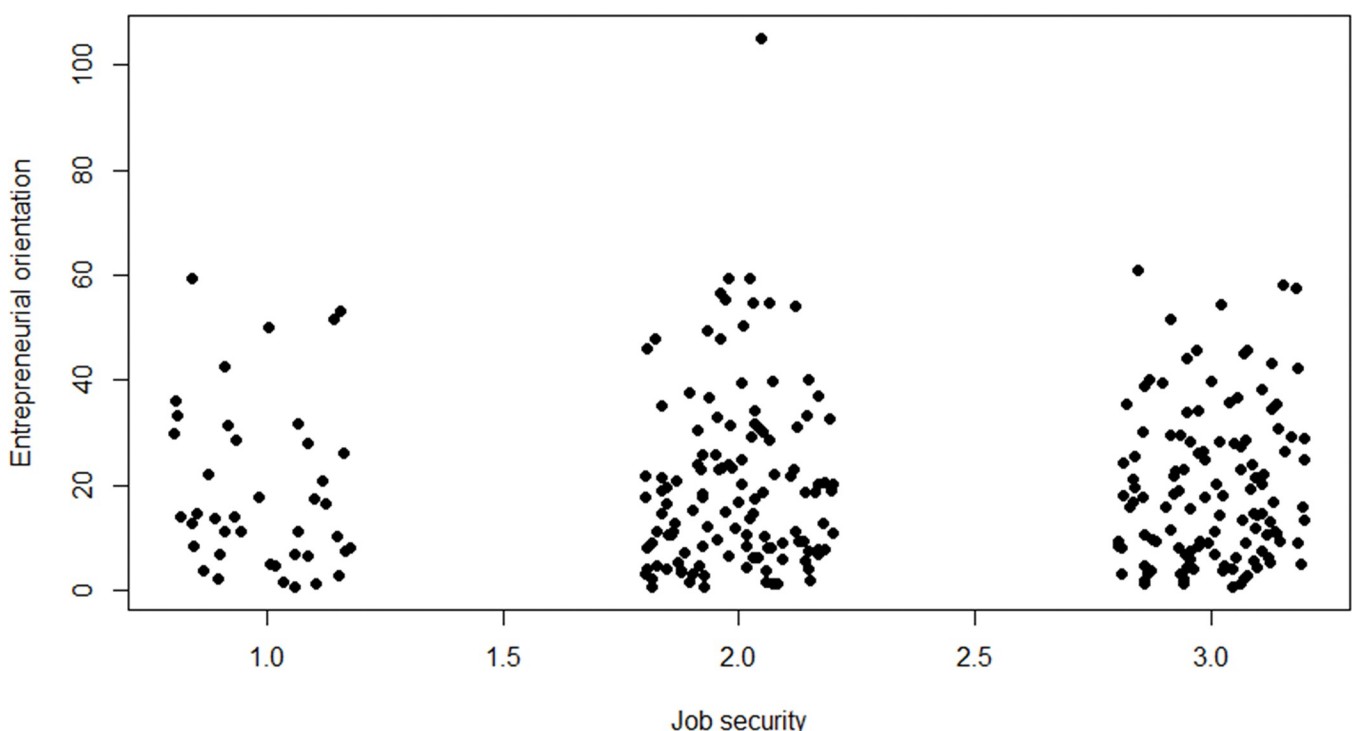

**Fig 8. Scatterplot of sense of job security and entrepreneurial orientation.** $n$ = 388; jitter (1) on sense of job security.

**Table 2. Main results (standardized effect sizes).**

| Type of hardship | Total effect on entrepreneurial orientation | Indirect effect on entrepreneurial orientation |
|---|---|---|
| Hard crisis times (number of employees) | -.09 | 9% via SJS (-.008)<br>3% via ESE (-.003) |
| Hard crisis times (profit) | -.07 | 20% via SJS (-.014)<br>10% via ESE (-.007) |
| Hard recovery times (number of employees) | -.17 | 4% via SJS (-.007)<br>7% via ESE (-.011) |
| Hard recovery times (profit) | -.05 | 42% via SJS (-.020)<br>33% via ESE (-.016) |

but ESE is the best of the rest. Regarding E2, Table 3 shows a consistently positive, but weak effect of SJS on EO: A 1 standard deviation increase in SJS is associated with a .05 to .07 standard deviation increase in EO.

Table 3 suggests that H3 is rejected because the sometimes weak H1 findings undermine the strong E1 results. Fig 9 visualizes that insight via a partial regression plot. Partial regression plots are like scatter plots, but they incorporate information of the control variables (see S1 File for details). We see the upward trend in the observation cloud in line with E1, like in Fig 7. However, we do not observe a clear trend of ventures with harder times having lower ESE (even for the type of hard times for which H1 had the strongest effect size): Ventures with neutral or better recovery times span the range of ESE, and ventures with very hard recovery times are not represented only in the region with extremely high ESE. Thus, the pattern is less clear than in the comparable plot in Fig 3, which does not account for the covariates.

Regarding H4, the strong H2 results are diluted by weak E2 findings. Fig 10 visualizes this insight. A majority of entrepreneurs with low SJS experienced very hard times, in line with Fig 5 and H2. However, the four observations with high SJS and EO residuals below -20 undermine a positive association between SJS and EO.

Fig 11 shows the partial regression plot for the type of hard times that has the strongest impact on EO–i.e., hard recovery times in terms of employees. We can immediately see that hard times of this type are less common than hard times in terms of profit. Possibly, entrepreneurs are less willing to let the number of employees fall below satisfaction than to let profit fall below satisfaction. The entrepreneurs care for their employees, finding it hard to let them suffer from the ventures' underperformance. Indeed, entrepreneurs may not only be motivated by the need for achievement, but also by the need for affiliation (e.g., [56]). An alternative explanation is that, entrepreneurs may be more flexible regarding the number of employees they want, and more rigid regarding the profit they want.

Harder times are relatively more common at lower levels of SJS (in line with H2). High EO is more common among entrepreneurs with greater SJS (in line with E2). Furthermore, in line with the main result, harder times are clearly overrepresented at the bottom of the figure (i.e.,

**Table 3. Evidence for the mediation paths overview (standardized effect sizes).**

| Type of hardship | Hard times on entrepreneurial self-efficacy | Hard times on sense of job security | Entrepreneurial self-efficacy on entrepreneurial orientation | Sense of job security on entrepreneurial orientation |
|---|---|---|---|---|
| Hard crisis times (number of employees) | -.02 | -.13 | .13 | .06 |
| Hard crisis times (profit) | -.05 | -.23 | .12 | .06 |
| Hard recovery times (number of employees) | -.10 | -.15 | .11 | .05 |
| Hard recovery times (profit) | -.13 | -.29 | .13 | .07 |

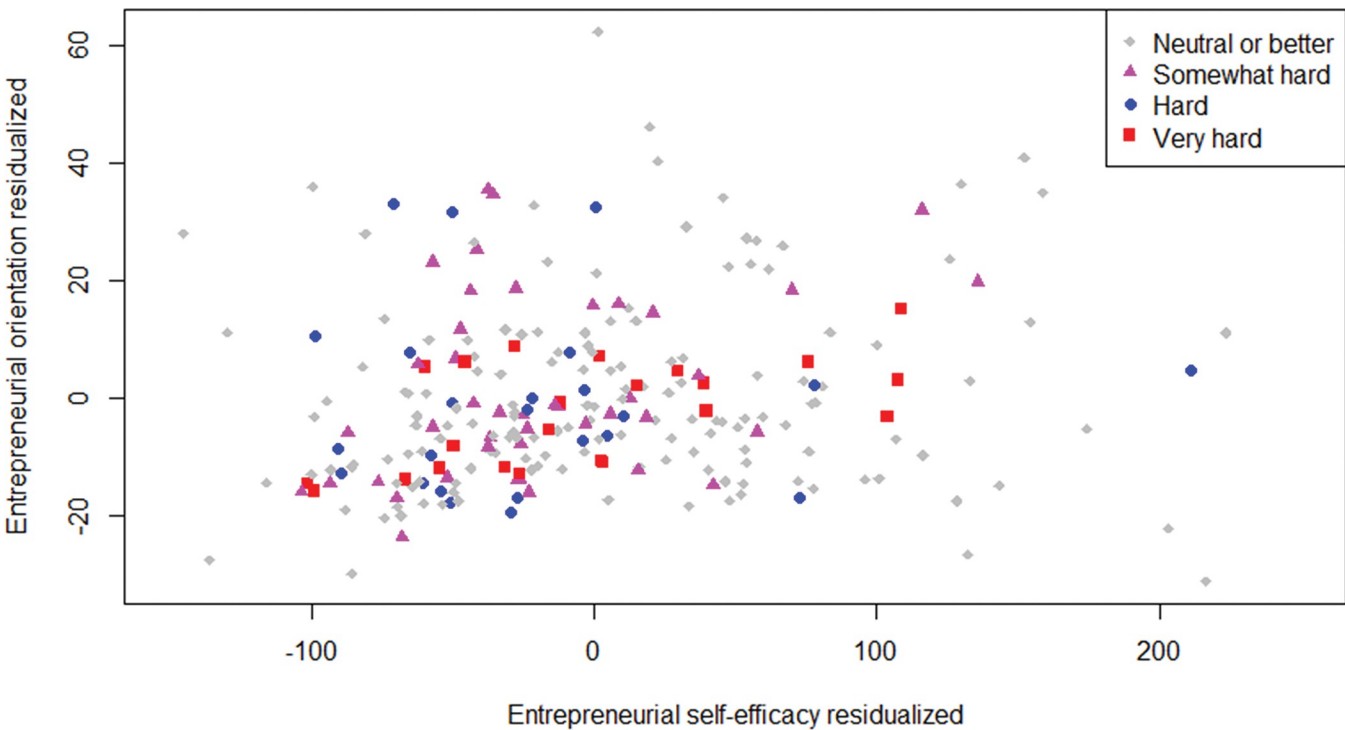

**Fig 9. Partial regression plot for ESE and EO.** *n* = 245; observation shape-color by levels of hard recovery times in terms of profit.

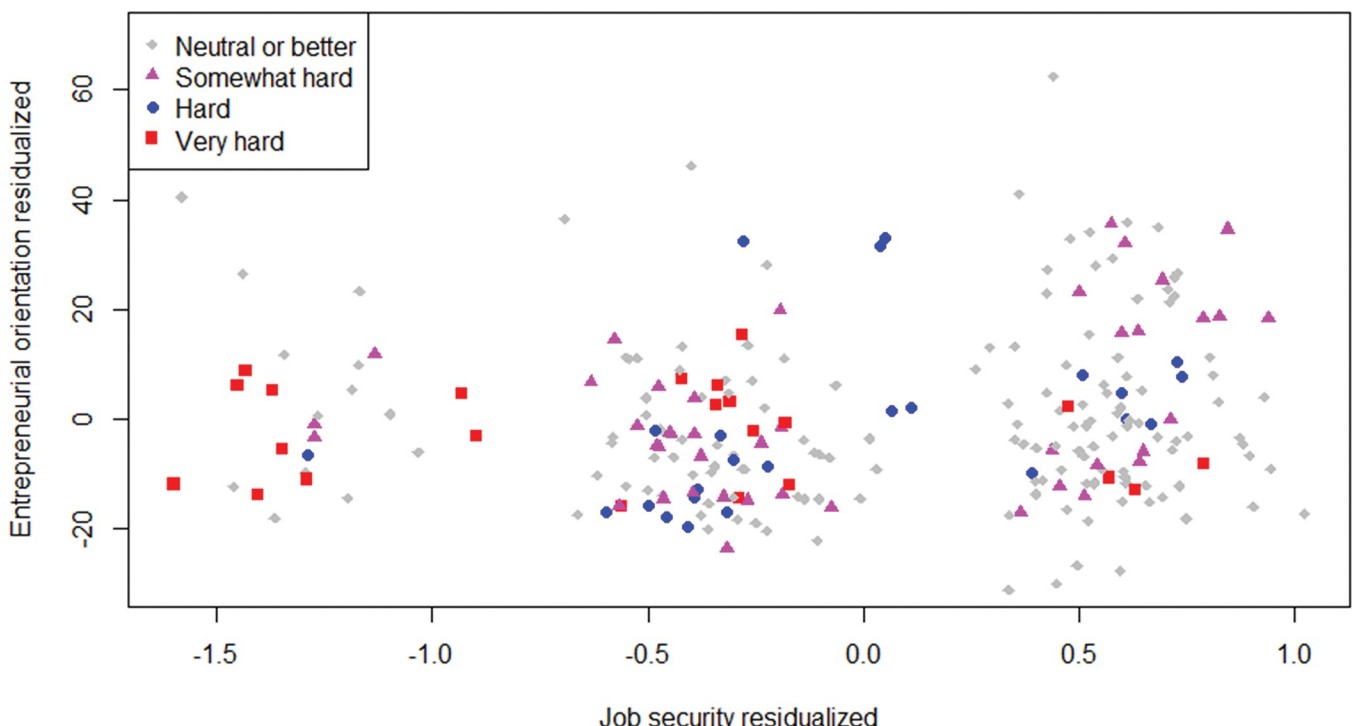

**Fig 10. Partial regression plot for SJS and EO.** *n* = 245; observation shape-color by levels of hard recovery times in terms of profit.

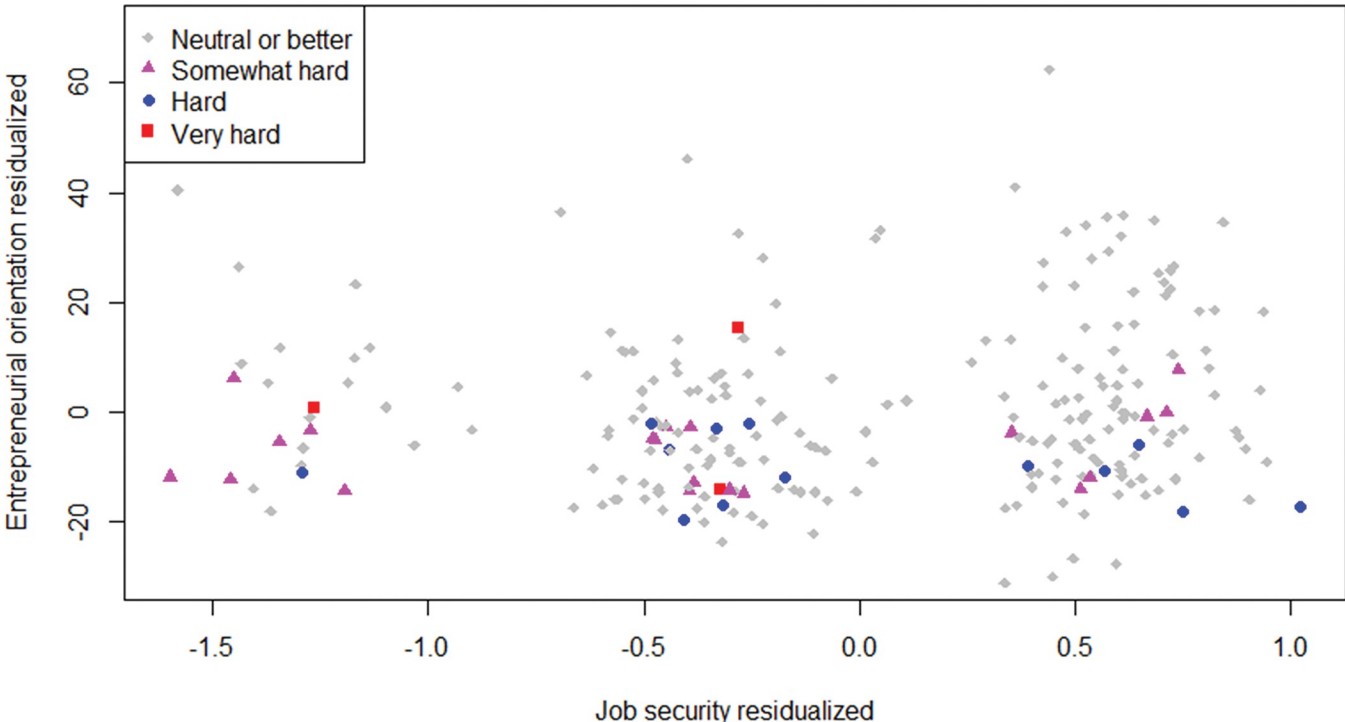

**Fig 11. Partial regression plot for SJS and EO with hard times in terms of employees.** $n$ = 245; observation shape-color by levels of hard recovery times in terms of employees.

the lower range of EO). In fact, for each level of SJS, entrepreneurs with hard times score lower on EO than entrepreneurs who experienced neutral or better recovery times. The fact that this pattern is similarly true across levels of SJS highlights the result that low SJS is not necessary for hard times to reduce EO. Specifically, if the few observations with hard times at high SJS had been spread across the range of EO, just like the neutral or better observations, the mediation would have been supported. But instead, the observations with hard times stick to the low EO range even at high SJS. In sum, hard times reduce the SJS, and this SJS slightly increases EO, and hard times reduce EO, but SJS does not mediate between hard times and EO.

Regarding H5 vis-à-vis H5alt, all results support H5 (our novel crisis imprinting theory) over H5alt (the behavioral theory of the firm). Table 2 shows that hard times has a consistent negative effect on EO, and its effect size is among the largest of the predictors in our data (see S1 File), especially for hard times in terms of the number of employees. One possibility to "save H5alt" would be that profit growth was a better measure of below-aspiration performance than hard times such that the behavioral theory's prediction would be observed for that variable instead. However, the profit trend shows a positive .10 to .12 standardized effect size across models. This also goes against the behavioral theory's prediction.

## Discussion

### Theoretical implications

This section considers the implications of our findings for theories that predict a relationship between hard times and EO. Specifically, in addition to CIT and the behavioral theory of the firm, for which we have developed hypotheses, we also discuss the trial by fire model. For CIT, we reflect on the weakness of the indirect effects of hard times on EO via ESE and SJS. First,

the indirect effect via ESE is weak due to the weak effect of some types of hard times on ESE. Hard times during crisis have a weaker effect on ESE than hard times after the crisis. This suggests the implication that our persistence premise may be weak for ESE. Second, the indirect effect via SJS is weak because, although hard times do reduce SJS, SJS is only weakly related to EO. A possible explanation for this is that the evidence from employee samples that SJS influences EO only partially applies to the entrepreneur context.

For the behavioral theory of the firm, we reflect on the lack of support its prediction four possible implications. What could explain the lack of support for the behavioral theory? First, It could be that profit growth was a better measure of below-aspiration performance than hard times such that the behavioral theory's prediction would be observed for that variable instead. However, the profit trend shows a positive .10 to .12 standardized effect size across models. This also goes against the behavioral theory's prediction.

Second, entrepreneurs let go of their original aspirations in a crisis such that low performance is no longer seen as below any aspiration. Such focus shift is usually discussed in terms of survival threat. In that case, ventures do not interpret their performance as below aspiration level, but as above survival level. That interpretation puts performance in a more positive light, which lowers motivation to change [57], consistent with what we find. There are two consideration that make this explanation seem less plausible. Specifcally, if ventures do not evaluate performance relative to aspiration, then that should also be reflected in their benchmark for judging satisfaction and thus our measure of hard times. Furthermore, salience is temporary by nature, so arguments involving salience are unlikely to explain the persistent effects we find.

A third intriguing possibility relates to the issue that theories tend to underspecify the timing of effects, as the social sciences tend to lack good theories on the duration and persistence of social processes. Here, the behavioral theory and CIT are no different. If theory cannot do the job, the alternative is to let the data speak. Our results show a risk-reduction effect after three to five years (i.e., this sample responded in 2017–2019 about the period after the crisis, which is approximately 2012–2014). The behavioral theory could be consistent with our result if it specified that below-aspiration performance increases risk-taking only in the next one or two years at the most (for SME samples, effect timing may be adjusted depending on firm size or R&D intensity). However, this speculation cannot explain why also the recent profit trend variable would have a positive coefficient.

Fourth, the implication may be thatbelow-aspiration performance theory does not generalize to small firms. That would also explain why the profit trend variable's coefficient is inconsistent with that theory. If so, the generalizability cutoff is between 50 and 250 employees, because our sample has only six ventures with more than 50 employees, and Osiyevskyy and Dewald [25] find results in line with below-aspiration performance arguments in a sample with mean firm size of around 250 employees. This is in line with findings in the context of new venture growth, where below-aspiration performance only increases growth at 55 or more employees [24]. If this interpretation is correct, there may be a boundary condition to CIT such that it applies only to smaller firms, while the behavioral theory of the firm holds for larger firms. Evidence on a largely negative effect of environmental hostility on EO among publicly listed firms [22], suggests CIT may also apply to larger firms. Of course, further work is needed to explore this intuition, and to precisely pin down any boundary conditions.

We would also like to reflect on the implications of our findings for the 'trial by fire' model [58]. The first aspect of that model is that hard times weed out the entrepreneurs with low performance. Similar to what we explained for survivorship bias, if low ESE and EO are associated with low performance, then the weeding out removes those observations. Therefore, only high ESE and EO would be left for researchers to observe among those with very hard times. The

second aspect is a causal mechanism, which is more developed under the banner of 'the red queen effect' [59]. The red queen effect says that strong competitive pressure (e.g., hard times) is associated with a success bias that increases confidence (e.g., SJS and ESE) and adaptation (e.g., EO) [60]. In sum, both aspects imply that hard times increase ESE and EO. However, we find a negative effect. What could explain that inconsistency of our findings vis-à-vis the 'trial by fire' and 'red queen' prediction? One speculation is that competitive pressure can be seen as a combination of (1) the size of the pie and (2) the number of actors that want a piece of this pie. Tests of trial by fire and red queen focus on variation in the number of competitors; the size of the pie is usually not specified. By contrast, the economic crisis was a case of a reduced pie size with a stable, if not decreasing, number of competitors. Thus, our findings may be inconsistent with the model because a reduction in the size of the pie is not what the model considers competitive pressure. If so, we contribute to work on the trial by fire model by identifying that a stable, if not growing, pie size is a crucial boundary condition for that model.

Another interpretation is that our data do not have a non-crisis comparison group. In that interpretation, the red queen effect causes the existence of ventures that experience neutral or better times during hard times. The model would predict that there are fewer such ventures in a non-crisis sample. The key question is whether SMEs that experience neutral or better times are (1) those SMEs within the sample with low competitive pressure or (2) those SMEs with positive outcomes given high competitive pressure for everyone in the sample. The latter answer means that our findings are inconsistent with the model because the model does not apply to our data. All this further strengthens our argument that we need more research on what hard vis-à-vis normal or prosperous times implies for the applicability of our theories, including the behavioral theory, trial by fire, red queen and our novel crisis imprinting theory. Preferably, we must collect longitudinal (panel) data with repeated measures across different 'types of times'.

## Limitations

The main goal of the study has been to assess the long term effects of a crisis. The main limitation of the study in that regard is that we did not measure the degree of hard times at the time of the crisis. We used covariates to attempt to exclude spuriousness this may cause in the form of hindsight bias, but future research ideally measures the degree of hard times contemporaneously and then follows up with those companies years later to assess their EO. Another advantages of that approach is that it allows the researcher to assess attrition, or 'survivorship bias'. As survivorship bias works against the CIT hypothesis, it is not an alternative explanation to our results. However, if future research can control for it, the results may be stronger.

Across types of hard times (i.e., employment versus profit), the results are consistent in terms of sign, but not in terms of effect size. The variation in effect size is difficult to explain, because different types of hard times are strongest for different outcome measures. For example, in Table 2, hard recovery times in terms of profit has the weakest effect, but in Table 3 it has the strongest effect. We find it hard to come up with a plausible explanation for this asymmetry. Clearly, we must conduct many more crisis-related studies with different SME samples to deepen our understanding of crises' multifaceted consequences, and develop fine-tuned theory accordingly.

## Conclusion

To advance the literature on SMEs in times of economic crisis [1–3, 6, 9, 10], particularly with respect to persistent effects for survivors of extreme adversity, we develop crisis imprinting theory (CIT). With that theory, we are able to examine how the crisis influences

entrepreneurial orientation, which adds insight to the small literature on the reversed performance-EO relationship. What we find, in many cases, is that hard times' effect size is meaningful when compared to other predictors such as profit growth. So, CIT appears to be useful for making accurate predictions about SME crisis resilience.

Practically, the effect of hard times on entrepreneurial orientation suggests that a vicious cycle in which hard times perpetuate themselves can be a real threat to the longer-run viability of the SMEs that have been able to survive economic adversity. Moreover, the mediators we studied explain only 11% of the largest effect, so SJS and ESE are highly incomplete as levers for increasing SME crisis resilience. In future work, therefore, we may examine interventions (e.g., post-crisis entrepreneurial coaching and training programs) that correct for the negative imprinting effect of going through hard times, as well as explore other mediators than the pair we gave central stage in the current paper. For example, some consequences of experiencing complete business failure [61] may, to a lesser extent, also result from hard times.

## Supporting information

**S1 File.**
(DOCX)

## Author Contributions

**Conceptualization:** Joeri van Hugten.

**Data curation:** Johanna Vanderstraeten, Wim Coreynen.

**Formal analysis:** Joeri van Hugten.

**Funding acquisition:** Johanna Vanderstraeten, Arjen van Witteloostuijn.

**Investigation:** Joeri van Hugten.

**Methodology:** Joeri van Hugten.

**Project administration:** Wim Coreynen.

**Resources:** Arjen van Witteloostuijn.

**Validation:** Wim Coreynen.

**Visualization:** Joeri van Hugten.

**Writing – original draft:** Joeri van Hugten.

**Writing – review & editing:** Johanna Vanderstraeten, Arjen van Witteloostuijn, Wim Coreynen.

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
