## [Decision Letter · Decision Letter 0]

13 Apr 2023

PONE-D-23-02387When the going gets tough, the entrepreneurs get less entrepreneurial?PLOS ONE

Dear Dr. van Hugten,

Thank you for submitting your manuscript to PLOS ONE. After careful consideration, we feel that it has merit but does not fully meet PLOS ONE’s publication criteria as it currently stands. Therefore, we invite you to submit a revised version of the manuscript that addresses the points raised during the review process. Please expand the originality and significance of contribution in introduction section by adding more arguments about the contribution to existing knowledge.IN the methods section there are some technical details that could improve readability of the paper. In conclusions you need to expand implications of your study and also add limitations which basically are missing.

We look forward to receiving your revised manuscript.

Kind regards,

Besnik Krasniqi, PhD

Academic Editor

PLOS ONE

Additional Editor Comments:

Dear Author(s)

We have received reviewers for you paper. The reviewers has recommended some minor revisions to your manuscript. Therefore, I invite you to respond to the reviewer(s)' comments and revise your manuscript.

Once again, thank you for submitting your manuscript to PLOS One

Reviewers' comments:

Reviewer's Responses to Questions

**Comments to the Author**

1. Is the manuscript technically sound, and do the data support the conclusions?

Reviewer #1: Yes

Reviewer #2: Yes

2. Has the statistical analysis been performed appropriately and rigorously? 

Reviewer #1: Yes

Reviewer #2: Yes

3. Have the authors made all data underlying the findings in their manuscript fully available?

Reviewer #1: Yes

Reviewer #2: Yes

4. Is the manuscript presented in an intelligible fashion and written in standard English?

Reviewer #1: No

Reviewer #2: Yes

5. Review Comments to the Author

Reviewer #1: METHOD

Table 1. Variables and measures show that rich data sources are used. This can be considered a strength of the research.

In line 274, we see that 5 companies in the sample have more than 50 employees. In the sample, you said that most of them are less than 5 employees. How many, please specify.

Sample. As part of an ongoing consulting project involving entrepreneurs and their SMEs, we asked 532 entrepreneurs in great detail about their venture, as an enterprise, and about themselves as an individual, a couple of weeks later. In the venture questionnaire, we measured EO and satisfaction with performance. In the individual survey, we measured our pair of mediators: ESE and SJS. Thus, measures of concepts directly linked in our conceptual model are never sourced from the same survey. This time lag substantially reduces the threat of common-method variance, probably to (close to) zero [49]. The sample contains entrepreneurs and their ventures from Flanders (the Dutch-speaking part of Belgium), from 55 three-digit industries. All but five ventures have less than 50 employees, and many even have less than five employees.

In line 388, we see that 1 company in the sample has more than 50 employees.

Sixth, larger ventures should be more buffered against hard times, so post-crisis recovery is less likely to be very dissatisfactory, and (sense of) job security should be greater. Very large ventures are often more inert, and thus less entrepreneurially oriented. However, all but one ventures in our sample have fewer than 50 employees. With that range, it may be that this covariate is not impactful.

Structural equation models estimated with the lavaan 396 packages in R were used in the research. Therefore, giving the script in Appendix H is essential for the reproducibility of the study.

RESULTS

Giving scatter diagrams as an attachment is healthier for reading the text. Otherwise, there are too many shapes. This can make the text difficult to read.

DISCUSSION

1-Main results are compatible with the crisis imprinting theory,

2- The behavioral theory of the firm is included in H5alt.

3- Another theory that may be related is the 'trial by fire' model

4- The second aspect is a causal 557 mechanism, which is more developed under the banner of "the red queen effect."

Including the first two theories in the introduction and literature and the third and fourth theories in the discussion creates confusion. However, when the discussion section is thoroughly read, it is seen that the justification is well done.

CONCLUSION

Please give abbreviations where they are first used: for example: "sense of job security (SJS)" on line 603. Please check all abbreviations in the article.

The indirect effects of hard times on EO via ESE and SJS are weak. The reasons for that weakness differs between them. The indirect effect via ESE is weak due to the weak effect of some types of hard times on ESE. Hard times during crisis have a weaker effect on ESE than hard times after the crisis, which suggests that our persistence premise may be weak for ESE. The indirect effect via sense of job security (SJS) is weak because, although hard times do reduce SJS, SJS is only weakly related to EO. A possible explanation for this is that the evidence from employee samples that SJS influences EO only partially applies to the entrepreneur context.

Please proofreading the entire article.

Reviewer #2: Thank you for providing to me the opportunity to review the article. I think that there are few studies that examine entrepreneurship during the crisis, thereby, this article makes contribution to the literature. There is no limitations and future suggestions section, I recommend to authors to write as sub-section within the discussion section.

6. PLOS authors have the option to publish the peer review history of their article (what does this mean?). If published, this will include your full peer review and any attached files.

Reviewer #1: No

Reviewer #2: No

---

## [Author Response · Author response to Decision Letter 0]

28 May 2023

Response to reviewers 

PONE-D-23-02387

When the going gets tough, the entrepreneurs get less entrepreneurial?

Thank you for the opportunity to revise and resubmit our paper. Please find our responses to your suggestions indented and in italics (in the word document version).

Editor’s comments:

Please expand the originality and significance of contribution in introduction section by adding more arguments about the contribution to existing knowledge.

 We added an explicit mention of the rich data we show an analysis of.

IN the methods section there are some technical details that could improve readability of the paper. 

 We have clarified the technical details where the reviewers noticed unclarity.

In conclusions you need to expand implications of your study and also add limitations which basically are missing.

We have clarified the implications in the conclusions by moving the more nuanced and involved discussion to the new limitations subsection in the discussion section. Now the conclusion really drives home the main message.

Furthermore, the discussion section is now reorganized into an implications subsection, where expanded implications are mentioned.

5. Review Comments to the Author

Reviewer #1: METHOD

Table 1. Variables and measures show that rich data sources are used. This can be considered a strength of the research.

 Thank you for appreciating the valuable and unique richness of our data. This comment inspired us to sell this richness explicitly in the introduction. 

In line 274, we see that 5 companies in the sample have more than 50 employees. In the sample, you said that most of them are less than 5 employees. How many, please specify.

Sample. As part of an ongoing consulting project involving entrepreneurs and their SMEs, we asked 532 entrepreneurs in great detail about their venture, as an enterprise, and about themselves as an individual, a couple of weeks later. In the venture questionnaire, we measured EO and satisfaction with performance. In the individual survey, we measured our pair of mediators: ESE and SJS. Thus, measures of concepts directly linked in our conceptual model are never sourced from the same survey. This time lag substantially reduces the threat of common-method variance, probably to (close to) zero [49]. The sample contains entrepreneurs and their ventures from Flanders (the Dutch-speaking part of Belgium), from 55 three-digit industries. All but five ventures have less than 50 employees, and many even have less than five employees.

In line 388, we see that 1 company in the sample has more than 50 employees.

Sixth, larger ventures should be more buffered against hard times, so post-crisis recovery is less likely to be very dissatisfactory, and (sense of) job security should be greater. Very large ventures are often more inert, and thus less entrepreneurially oriented. However, all but one ventures in our sample have fewer than 50 employees. With that range, it may be that this covariate is not impactful.

We have corrected this inconsistency (also in line 578). There are 5 companies with between 50 and 100 employees, and one company with 100-200 employees.

We now also specify the number of companies with less than 5 employees (104).

Structural equation models estimated with the lavaan 396 packages in R were used in the research. Therefore, giving the script in Appendix H is essential for the reproducibility of the study.

 Thank you for appreciating our efforts to make the study as reproducible as we can.

RESULTS

Giving scatter diagrams as an attachment is healthier for reading the text. Otherwise, there are too many shapes. This can make the text difficult to read.

We agree with this concern but propose a different solution. Our plan is to ensure readability at the copyedit stage by indicating that the scatter diagrams should be published at half the size of regular figures. But we are open to other suggestions.

DISCUSSION

1-Main results are compatible with the crisis imprinting theory,

2- The behavioral theory of the firm is included in H5alt.

3- Another theory that may be related is the 'trial by fire' model

4- The second aspect is a causal mechanism, which is more developed under the banner of "the red queen effect."

Including the first two theories in the introduction and literature and the third and fourth theories in the discussion creates confusion. However, when the discussion section is thoroughly read, it is seen that the justification is well done.

Thank you for appreciating the justification, and thank you for reading thoroughly. We reorganized the discussion to reduce confusion.

CONCLUSION

Please give abbreviations where they are first used: for example: "sense of job security (SJS)" on line 603. Please check all abbreviations in the article.

The indirect effects of hard times on EO via ESE and SJS are weak. The reasons for that weakness differs between them. The indirect effect via ESE is weak due to the weak effect of some types of hard times on ESE. Hard times during crisis have a weaker effect on ESE than hard times after the crisis, which suggests that our persistence premise may be weak for ESE. The indirect effect via sense of job security (SJS) is weak because, although hard times do reduce SJS, SJS is only weakly related to EO. A possible explanation for this is that the evidence from employee samples that SJS influences EO only partially applies to the entrepreneur context.

Thank you for pointing this out. We have corrected this and also doublechecked all other abbreviations.

Please proofreading the entire article.

We carefully doublechecked the article for any spelling or grammar issues.

Reviewer #2: Thank you for providing to me the opportunity to review the article. I think that there are few studies that examine entrepreneurship during the crisis, thereby, this article makes contribution to the literature. There is no limitations and future suggestions section, I recommend to authors to write as sub-section within the discussion section.

Thank you for recognizing our contribution to the literature. We have added a limitations and future research subsection as recommended. This comment pushed us to reorganize our discussion and conclusion into a more focused format, improving the paper.

 We have paid extra attention to being as specific as possible about our data availability.

---

## [Decision Letter · Decision Letter 1]

4 Aug 2023

When the going gets tough, the entrepreneurs get less entrepreneurial?

PONE-D-23-02387R1

Dear Dr. Joeri van Hugten,

We’re pleased to inform you that your manuscript has been judged scientifically suitable for publication and will be formally accepted for publication once it meets all outstanding technical requirements.

Kind regards,

Abdullah Al Mamun, PhD

Academic Editor

PLOS ONE

Additional Editor Comments (optional):

Reviewers' comments:

Reviewer's Responses to Questions

**Comments to the Author**

1. If the authors have adequately addressed your comments raised in a previous round of review and you feel that this manuscript is now acceptable for publication, you may indicate that here to bypass the “Comments to the Author” section, enter your conflict of interest statement in the “Confidential to Editor” section, and submit your "Accept" recommendation.

Reviewer #1: All comments have been addressed

2. Is the manuscript technically sound, and do the data support the conclusions?

Reviewer #1: Yes

3. Has the statistical analysis been performed appropriately and rigorously? 

Reviewer #1: Yes

4. Have the authors made all data underlying the findings in their manuscript fully available?

Reviewer #1: Yes

5. Is the manuscript presented in an intelligible fashion and written in standard English?

Reviewer #1: Yes

6. Review Comments to the Author

Reviewer #1: (No Response)

7. PLOS authors have the option to publish the peer review history of their article (what does this mean?). If published, this will include your full peer review and any attached files.

Reviewer #1: No

---

## [Editor Report · Acceptance letter]

14 Aug 2023

PONE-D-23-02387R1 

When the going gets tough, the entrepreneurs get less entrepreneurial? 

Dear Dr. van Hugten:

I'm pleased to inform you that your manuscript has been deemed suitable for publication in PLOS ONE. Congratulations! Your manuscript is now with our production department. 

Kind regards, 

on behalf of

Assoc. Prof. Dr. Abdullah Al Mamun 

Academic Editor

PLOS ONE